# Enhanced Arabic Sentiment Analysis Using a Novel Stacking Ensemble of Hybrid and Deep Learning Models

Hager Saleh [1,*], Sherif Mostafa [1], Lubna Abdelkareim Gabralla [2], Ahmad O. Aseeri [3,*] and Shaker El-Sappagh [4,5]

1 Faculty of Computers and Artificial Intelligence, South Valley University, Hurghada 1974531, Egypt
2 Department of Computer Science and Information Technology, College of Applied, Princess Nourah Bint Abdulrahman University, P.O. Box 84428, Riyadh 11671, Saudi Arabia
3 Department of Computer Science, College of Computer Engineering and Sciences, Prince Sattam Bin Abdulaziz University, Al-Kharj 11942, Saudi Arabia
4 Faculty of Computer Science and Engineering, Galala University, Suez 435611, Egypt
5 Information Systems Department, Faculty of Computers and Artificial Intelligence, Benha University, Banha 13518, Egypt
* Correspondence: hager.saleh@fcih.svu.edu.eg (H.S.); a.aseeri@psau.edu.sa (A.O.A.)

**Abstract:** Sentiment analysis (SA) is a machine learning application that drives people's opinions from text using natural language processing (NLP) techniques. Implementing Arabic SA is challenging for many reasons, including equivocation, numerous dialects, lack of resources, morphological diversity, lack of contextual information, and hiding of sentiment terms in the implicit text. Deep learning models such as convolutional neural networks (CNN) and long short-term memory (LSTM) have significantly improved in the Arabic SA domain. Hybrid models based on CNN combined with long short-term memory (LSTM) or gated recurrent unit (GRU) have further improved the performance of single DL models. In addition, the ensemble of deep learning models, especially stacking ensembles, is expected to increase the robustness and accuracy of the previous DL models. In this paper, we proposed a stacking ensemble model that combined the prediction power of CNN and hybrid deep learning models to predict Arabic sentiment accurately. The stacking ensemble algorithm has two main phases. Three DL models were optimized in the first phase, including deep CNN, hybrid CNN-LSTM, and hybrid CNN-GRU. In the second phase, these three separate pre-trained models' outputs were integrated with a support vector machine (SVM) meta-learner. To extract features for DL models, the continuous bag of words (CBOW) and the skip-gram models with 300 dimensions of the word embedding were used. Arabic health services datasets (Main-AHS and Sub-AHS) and the Arabic sentiment tweets dataset were used to train and test the models (ASTD). A number of well-known deep learning models, including DeepCNN, hybrid CNN-LSTM, hybrid CNN-GRU, and conventional ML algorithms, have been used to compare the performance of the proposed ensemble model. We discovered that the proposed deep stacking model achieved the best performance compared to the previous models. Based on the CBOW word embedding, the proposed model achieved the highest accuracy of 92.12%, 95.81%, and 81.4% for Main-AHS, Sub-AHS, and ASTD datasets, respectively.

**Keywords:** machine learning; deep learning; ensemble learning; Arabic sentiment analysis

## 1. Introduction

Social media services such as Facebook, Twitter, LinkedIn, and others have grown exponentially over the last decade. Companies and organizations have discovered that these platforms may be a great source of information for interacting with and learning more about their customers. However, quantifying a user's overall enjoyment of a brand can be extremely difficult due to the large number of users, posts, comments, messages, and other forms of contact [1]. Sentiment analysis is a subfield of natural language processing

(NLP) in which advanced data mining and machine learning models are used to measure sentiments, emotional responses, and attitudes in a variety of domains, such as service quality, product acceptance, price trends, and popular support of government actions and events [2]. Arabic has several irregular forms, intricate morpho-syntactic agreement rules, and a variety of linguistic varieties with no established writing standards. Learning stable general models over Arabic text may be difficult without suitable processing and handling. In terms of Sentiment Lexicons and Annotated Sentiment Corpora, Arabic Sentiment Analysis has fewer resources than English Sentiment Analysis does. Arabic Sentiment Analysis has garnered a lot of interest due to these challenges. (ASA) [3].

Deep learning (DL) and machine learning (ML) techniques can provide an automated mechanism for processing and extracting valuable data and sentiments from enormous amounts of text, and [1]. Recently, convolutional neural network (CNN) models and hybrid models of CNN and other DL models, such as long short-term memory (LSTM), provided significant improvements in performance in sentiment analysis [4–6]. CNN uses deep layers such as convolutional, pooling, and fully connected layers to extract more profound and essential features from text data. LSTM has a memory state that effectively memorizes necessary information in the text and understands the meaning of the whole sentence. For instance, Al Omari et al. [7] proposed a hybrid CNN-LSTM model employing word2vec word embeddings for the binary classification of Arabic attitudes, combining the strengths of both CNN and LSTM. Alwehaibi et al. [8] proposed a hybrid LSTM-RNN model based on LSTM and recurrent neural network (RNN) to analyze Arabic sentiment. The performance of the hybrid DL models can be improved by integrating more than one model [4], constituting an ensemble. An ensemble classifier is formed by combining the results of several classifiers to allow component models to balance out each other's shortcomings. In the literature on machine learning, ensemble learning techniques are receiving more attention. However, its usage in sentiment analysis is still limited, especially for the current literature on ensemble learning, which concentrates on homogeneous ensembles. Still, heterogeneous ensembles based on different base classifiers and datasets are expected to enhance the performance of the resulting ensembles. Recently, ensemble modeling has been used as a popular technique to boost the performance of NLP models [9]. Ensemble classifiers integrate the decisions of multiple classifiers, and the combined version of the resulting model is expected to improve the results of each base classifier [10]. In other words, ensemble allows learning models to alter the final ensemble model's weights for each base NLP system to optimize the whole model's decisions. However, in the absence of sufficient data, this training-based ensemble is prone to overfitting. Whether predictions and base learners are combined using meta-learning or rule-based approaches, as well as whether the learning process is carried out sequentially or concurrently, have a substantial impact on the ensemble learning process [11,12]. Heterogeneous ensembles consist of several classifiers, whereas homogeneous ensembles use repeated examples from the same base model. Different techniques are used to increase the variance among base classifiers in both homogeneous and heterogeneous ensembles. Heterogeneous ensembles have many types of bias, where the combination of these biased decisions could outperform the homogeneous ensembles if these prejudices are mutually beneficial [13]. In most circumstances, ensemble learning methods can take one of three forms: bagging [14], boosting [15], or stacking [16].

Therefore, we have proposed an optimized heterogeneous ensemble stacking model based on the best combination of CNN, hybrid CNN-LSTM, and CNN-GRU for Arabic sentiment analysis with SVM as a meta-learner. After optimization of the resulting model, it registered the best performance compared with other models.

Our contributions can be summarized as follows:

- We proposed three DL architectures: deep convolutional neural network (DeepCNN), hybrid CNN-LSTM, and hybrid CNN-gated recurrent unit (GRU). A Bayesian optimizer has been used to optimize the hyperparameters of these DL models.
- We proposed a heterogeneous ensemble stacking model that combined the three pretrained DL models of DeepCNN, hybrid CNN-LSTM, and hybrid CNN-GRU. SVM has been used as the meta-learner to combine the outputs of the three base DL models.
- To evaluate the superiority of the proposed ensemble model, the performance of the stacking model has been compared with the performance of several DL and classical ML models using the three well-known Arabic datasets of Arabic health services datasets (Main-AHS and Sub-AHS) and the Arabic sentiment tweets dataset (ASTD).
- The proposed ensemble stacking model significantly outperformed other deep learning models in terms of accuracy, precision, recall, and f1-score.

Our paper is organized as follows. The associated sentiment analysis for Arabic models relevant to our work is described in Section 2. Section 3 summarizes the proposed strategy and presents the proposed model. The experimental results are summarized and discussed in Section 4. Finally, Section 5 concludes the paper.

## 2. Related Work

DL has recently demonstrated its effectiveness compared to the state-of-the-art performance of standard ML methods for sentiment analysis. To identify sentiments in SemEval 2017, ASTD, and ARSAS, the authors in [17] utilized a CNN and LSTM. They generated a word embedding matrix using word2vec. Their model achieved the highest performance for SemEval 2017 and ASTD datasets. In [7], for the binary classification of Arabic attitudes, the authors suggested a hybrid CNN-LSTM model with word2vec embeddings. They used many Arabic sentiment analysis datasets, such as Main-AHS, Ar-Twitter, and ASTD. The hybrid CNN–LSTM model registered the highest accuracy at 79.07%. The authors of [18] applied CNN on nine datasets, including LABR and ASTD, to analyze a binary sentiment analysis. The dataset includes two domains: reviews and tweets. The two types of word2vec of CBOW and Skip-Gram were used to generate the word embedding matrix. Additionally, they used CNN on both balanced and unbalanced datasets. In [8], the authors proposed a hybrid LSTM-RNN model based on LSTM and RNN to analyze Arabic sentiment. They studied the effect of using different pre-trained word embeddings with DL models. They tested their model with the AraSenTi-Tweet. In [19], the authors introduced an Arabic language dataset about health services from Twitter. They added annotations to tweets and pre-processed them into good and negative tweets. The authors applied NB, SVM, LR, and CNN to the health dataset. Using character-level data, the authors [20] deployed deep CNNs for Arabic sentiment analysis. The proposed large dataset was built on the available Arabic sentiment analysis datasets in different domains (modern standard, dialectal) to train networks. In addition, different ML algorithms, such as LR, SVM, and NB, have been applied to assess the performance on a large dataset. The results show that deep CNNs registered the highest accuracy compared to ML classifiers.

In [21], the authors investigated the bidirectional LSTM network (BiLSTM) to analyze Arabic sentiment analysis. Six Arabic datasets were used to train and evaluate their proposed model, the DL model and the ML model. Their proposed model achieved the highest performance compared to the performance of DL and ML models. In [22], the authors applied different ML algorithms and CNNs models with other feature extraction methods. They used Main-AHS Sub-AHS heath Arabic datasets. The result showed that CNN improved accuracy from 91 to 95% for publicly available Arabic language health sentiment datasets.

Ensemble models could enhance the inference power of single models. In addition, using hybrid models as base classifiers in an ensemble could boost the performance of hybrid models [9]. Ensemble models have been applied in different domains and achieved better results than base models [23]. In the Arabic sentiment analysis domain, ensemble modeling has been applied, such as in [9], where the authors proposed an ensemble model using voting for optimizing the Arabic sentiment analysis. They applied CNN-LSTM and the optimization method to select the best CNN and LSTM on the Arabic sentiment tweets dataset (ASTD). The selected models are the ones achieving the highest f1-score among other models. The authors of [4] proposed an Arabic sentiment dataset about COVID-19-related conspiracy theories. The collected data were annotated into positive and negative class labels. They applied RF, SVM, NB, and LR to the collected dataset and the SMOTE technique to handle unbalanced data. Furthermore, they used the voting classifier combining RF, SGD, SVM, BNB, and LR. The experimental results showed that applying the ensemble model with SMOTE improved the performance. In [22], the authors studied the effect of combining multi models (NB, SVM, and maximum entropy) using the voting algorithm on Arabic sentiment analysis. They applied models to different stemming, such as Khoja, ISRI, Tashaphyne, Light10, and MOTAZ. The outcomes demonstrated that the voting algorithm delivered the best results. In [9], the authors described an ensemble model based on CNN and LSTM that might be used to forecast the sentiment of Arabic tweets. To obtain experimental findings, they made use of the ASTD dataset. According to the results, the ensemble model has the best accuracy and f1-score.

Prior research employed voting ensemble learning models, hybrid models, and conventional ML. Furthermore, we proposed a stacking ensemble model in a previous study [10]. In that framework, we studied the role of the RNN family, including RNN, LSTM, and GRU, in interpreting the text data. On the other side, in the current study, we extend the previous work by adding new features. We explore the role of increasing the diversity of base classifiers by integrating CNN, LSTM, and GRU, which have extremely different learning philosophies. In addition, we further explore the role of combining CNN and LSTM in a hybrid model and CNN and GRU in a hybrid model and use these hybrid models as a base classifier in the stacking architecture. We expect that the heterogeneity of the base classifiers and the deep features extracted from hybrid models based on CNN is able to enhance the learning capabilities of the resulting ensemble model.

In [24], the authors analyzed the effect of inverters on sentiment analysis of Facebook dialectal Arabic posts. Furthermore, they studied the effect of negating words on the sentiment polarity of a post. They used F1-score, precision, and recall to evaluate their work. In our work, we applied ML classification algorithms, DL algorithms, hybrid DL algorithms, and the proposed ensemble stacking models to Arabic tweets data to classify sentiment analysis as positive or negative. We used TF-IDF and word embedding to extract features. We evaluated models using different evaluation metrics, accuracy, precision, recall, f1-score, and AUC.

## 3. Methodology

The suggested ensemble model is described in this section. It represents the essential phases in Arabic language sentiment analysis prediction. The proposed framework's general phases, comprising data pre-treatment, feature extraction, optimization, and classification steps, are depicted in Figure 1.

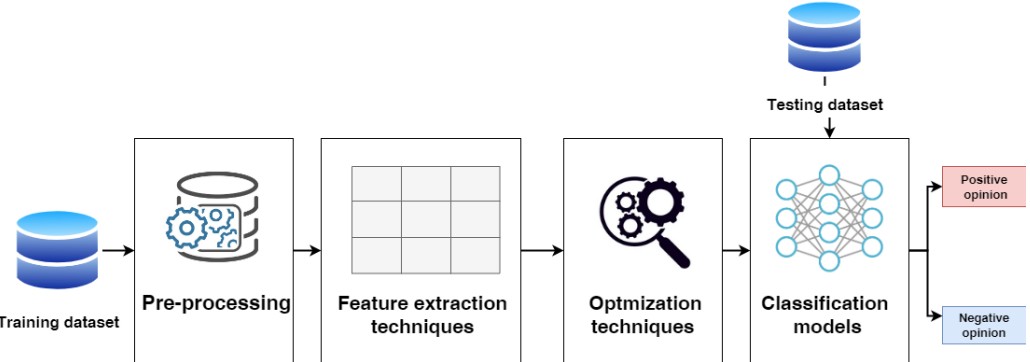

**Figure 1.** The main phases of prediction Arabic sentiment analysis.

*3.1. Data Pre-Processing*

The following pre-processing steps are used to prepare the datasets:

- Cleaning Tweets: this step includes removing HTML tags, URLs, and non-Arabic characters.
- Tokenizing: this step divides the text into parts.
- Pre-processing by removing stop words is a critical step in text pre-processing of sentiment analysis [25,26] because stop words are a collection of words that do not change the meaning of the text or do not hold information, such as prepositions, conjunctions, and articles. Furthermore, they are used to eliminate unimportant words, allowing algorithms to focus on the important words instead. We remove stop words using a stop words list, for example, some words. حيث ــ لدى ــ الا ــ عن ــ بِ إلى ــ لنا ــ فقط ــ الذي

ــ ذلك ــ مثل

- Stemming: the main job of the stemmer is to return the word to its root.
- Removing emojis.

*3.2. Feature Extraction*

- For classical ML models, the term frequency/inverse document frequency with N-gram is used to build the feature matrix.

    - N-grams are commonly employed in text mining and natural language processing. They are essentially a group of co-occurring words within a particular frame, and computing the n-grams normally moves one word forward, while in more complex cases, it can move N-words forward. N-grams are utilized to keep the context of newly acquired words. It employs a collection of sequentially ordered words based on the value of the N variable. If N = 1, it might be a unigram, and a bigram if N = 2 [27].
    - TF-IDF is a statistical measure used to weigh the importance of each word in the corpus. It is a feature extraction method used for classification and recommendation in NLP. The TF-IDF implementation process is divided into two parts. Begin by counting how many times each term appears in the document or tweet. The frequency of each word occurrence (IDF) was then calculated over all papers or tweets. The less important the term, the lower the TF-IDF value. The bigger TF-IDF values, on the other hand, indicate that there are fewer common words in the corpus and therefore are significant [28,29].

- For DL models, word embedding is used to present the word matrix. Word embedding is a method for transforming words in textual input into vectors. It is superior to traditional bag-of-words encoding techniques, which use enormous sparse vectors to encode a whole vocabulary by scoring each word in a vector. The basic idea behind word embedding is that words similar to each other will be adjacent in space. The word's "embedding" refers to its location in the learned vector space [30]. Word

embedding can also be taught as part of a deep learning model, which takes longer but can be customized to a specific training dataset. Each word is represented as a multidimensional unique feature vector in the vector space of a selected dimension in a word embedding. The fundamental idea is to put feature vectors for frequently occurring words in close proximity in space [31]. We used the AraVec word embedding, which is a Python-based open-source project that seeks to provide robust and free word embedding models to the Arabic NLP research community through the usage of the pre-trained distributed word representation. Words are represented in a continuous space as vectors, with numerous syntactic and semantic links encoded between them [32,33]. We used two approaches of AraVec, which are the CBOW and the skip-gram models with 300 dimensions. The CBOW model learns embeddings by predicting the middle word in a sequence based on the words in that sequence, regardless of their order in the sentence. The skip-gram Model seeks to predict the surrounding contextual words given the core word.

### 3.3. Hyperparameter Optimization

Hyperparameter optimization is the process of determining the optimal collection of values for hyperparameters for ML and DL models. ML algorithms are improved by grid search and cross-validation, while a Bayesian optimizer is used to optimize the architecture and the hyperparameters of DL models. We use the standard KerasTuner implementation for the Bayesian optimizer.

- Grid Search: The hyperparameters' ideal values are obtained via a tuning procedure. When the Model had several hyperparameters, it became required to search in a multi-dimensional space for the best combination of values for the hyperparameters [34,35]. Grid search is a hyperparameter tuning method that divides the hyperparameter domain into distinct grids and obtains the optimal combination of hyperparameter values.

- An evaluation method for learning algorithms known as cross-validation separates data into two parts: one for training models while the other is for model verification [36]. Cross-validation includes a single parameter, k, that indicates how many groups a given data sample should be divided into, which is why it is also known as k-fold cross-validation. Cross-validation is seen to be a strong preventive technique against overfitting because the first fold is utilized for the validation set, while the other k-1 folds are provided to the learning system to ensure that the model is estimated using data that were not seen during training [37].

- The KerasTuner hyperparameter optimization system includes the hyperband, Bayesian optimization, and random search algorithms. The optimal hyperparameter values for the models are found by employing one of the search algorithms after the search space is set up using a define-by-run syntax [38]. Hyperparameters are variables that govern the model's topology and training process and remain constant throughout the training phase, affecting the model's performance. There are two sorts of hyperparameters: process hyperparameters, which impact the quality and speed of the learning algorithm, and model hyperparameters, which control the number and breadth of hidden layers in the model [39]. For sophisticated models, the number of hyperparameters can be considerably expanded, making manual tuning difficult, and underscoring the need for the techniques. In our work, we optimized some of the values of parameters set for DeepCNN, CNN-LSTM, and CNN-GRU, as shown in Table 1.

**Table 1.** Range of possible values for hyperparameters of DeepCNN, CNN-LSTM, and CNN-GRU DL models.

| Parameters | Values |
| --- | --- |
| Num_filters | [64, 128, 256, 512] |
| Kernel_size | [2, 3, 4, 5] |
| Pool_Size | [2, 3, 4, 5] |
| LSTM_Unit | Range (50, 1000) |
| GRU_Unit | Range (50, 1000) |
| Dense_Unit | Range (50, 1000) |
| learning_rate | Between $1 \times 10^{-2}$ and $1 \times 10^{-7}$ |

*3.4. Machine Learning Algorithms*

We used different ML algorithms with TF-IDF, unigram, and bigram. ML algorithms are described in the following discussion.

- Logistic regression (LR): Predictions using logistic regression result in discrete values that are best suited for binary categorization. When determining whether or not an event will occur, there are only two options: it will occur or not occur in the binary classifications or even in multi-class classification, and the threshold has to be always specified to distinguish between them [40]. The logistic function (transformation function) or logistic curve (sigmoid curve) is a typical S-shaped curve with the equation, where $x$ is the sigmoid's midpoint, $L$ is the curve's peak value, and $k$ is the steepness of the curve or the logistic growth rate. The usual name for the typical logistic function is the sigmoid, which has $L = 1$, $k = 1$, and $x_0$.

$$f(x) = \frac{L}{1 + e^{-k(x - x_0)}} \tag{1}$$

As a result, an S-curve emerges. The default class is given a set of probabilities as a result of logistic regression different from linear regression, where output is produced straight away. The outcome is between 0 and 1 because it is a probability. The y-value is derived by log converting the x-value

$$f(x) = \frac{1}{1 + e^{-x}} \tag{2}$$

using the logistic function [41]. After that, a threshold is used to turn the probability into a binary category.

- Naïve Bayes (NB) determines whether the existence of a particular feature in a given class is independent of the existence of any other feature. The Bayes Theorem is used to find the probability of an event occurring in the case that the other event has already occurred [42]. Given our prior knowledge (*d*), Bayes' theorem is used to assess the probability of a hypothesis (h) being true:

$$P(b \mid d) = (P(d \mid h)P(h))/P(d) \tag{3}$$

where $P(h/d)$ denotes past probability. $P(d/h)$ = likelihood of data $d$ given data $dP(h)$ is the prior probability of a class if hypothesis $h$ is true, the chance that hypothesis "$h$" is right (regardless of the evidence), and $P(d)$ is the predictor's prior probability. Probability of the data (irrespective of the hypothesis). The multinomial model and Bernoulli model are two distinct techniques to build up Naive Bayes [43]. The documents are the classes that are handled as a separate "language" in the multinomial model's estimate. Bernoulli-NB (Bernoulli Naive Bayes) is a discrete data model that works with occurrence counts and is designed for Boolean/binary characteristics.

- Random forest (RF) takes a group of weak learners and combines them to build a stronger classification predictor [44]. The random forest tree's main purpose is to use a learning algorithm to merge numerous base-level predictors into a single effective and resilient predictor. In order to classify a new object relying on its attributes, each tree gave a classification and said that the tree "votes" for that class [45]. In the case of regression, the forest picks the category with the most votes (across all forest trees) and takes the average of outputs from different trees. The forest classifiers are fitted using two arrays, one with training data and the other with the goal values of the testing data while creating the random forest tree [46].
- A subset of the supervised learning algorithm family is the Decision Tree (DT) algorithm. A DT is used to develop a training model that can predict the class or value of the target variable by learning fundamental decision rules from prior data, which are the training data. DT has a tree shape structure with internal branches that represent the test outcome, while the leaves or terminal nodes have the class label. The source set is divided into subgroups using an attribute value test, and the result is a trained tree. Recursive partitioning is the process of repeating this action for each derived subset [47].
- KNN is a Supervised ML algorithm [48] which classify data based on similarity through comparing the new case or set of data to the existing cases to place it in the category that is most like the existing categories [49]. As a result, the KNN can swiftly classify new data as it comes in. KNN saves the dataset and then executes an action on it during classification instead of immediately learning from the training set. The KNN approach saves the dataset during the training phase and, when new data is received, classifies it into a category based on the Euclidean distance of the K number of neighbors highly similar to the new data [50].

### 3.5. Deep Learning Algorithms

We proposed three DL models, deep convolutional neural networks (DeepCNN), hybrid CNN-LSTM based on CNN and LSTM, and hybrid CNN-GRU based on CNN and GRU. The architectures of the three models are shown in Figure 2. Each model is described in detail as follows.

- The DeepCNN model includes different layers: the embedding layer, three CNN layers, two MaxPooling layers, Global MaxPooling, flatten, fully connected, and output layers.

  - The first layer utilized the embedding layer as a pre-processing step to turn the vector representation into a fixed-sized denser vector representation [51]. It is implemented in the Keras library [52]. Input-dim, output-dim, and input-length are the three parameters that are employed. Input-dim provides the vocabulary size of the dataset, output-dim describes the vector space in which words will be embedded, and input-length describes the length of input sequences. Because both the CBOW and the skip-gram Model are 300 d vectors long, we set the output-dim to 300, the input-dim to 20,000, and the input length to 140.
  - The multi-layer neural network known as convolutional neural networks (CNN) is an enhancement of the error backpropagation network [53]. It has a feature map and a convolution filter (kernel). The convolution filter is applied to the input word matrix to create a feature map identifying significant input data patterns. The base of the convolutional operation is the kernel function. Feature extraction is completed by sliding the kernel from top to bottom and from left to right in the input matrix. Each filter also makes use of the rectified linear unit (ReLU) activation function [54] to recognize various aspects of the news.
  - The max-pooling layer down samples the feature maps to be more resilient to the probable changes of a feature's position in the text by recapping the feature's presence in patches of the feature map. Calculate the maximum value for each feature map patch.

- The two-dimensional arrays of the combined feature maps are flattened into a single, lengthy continuous linear vector using the flatten layer [51].
- The fully connected layer is a dense and deeply connected layer compared to its preceding layer and changes the dimension of the output by implementing a vector-matrix multiplication.
- The output layer uses the output of the fully linked layer to determine if the tweets are positive or negative. In this layer, the ADAM optimizer [55] was employed, and the activation function is sigmoid [56].

- The embedding layer, CNN and MaxPooling layers, long short-term memory (LSTM), fully linked layer, and output layer are all components that constitute the hybrid CNN-LSTM model.

  LSTM is a recurrent neural network dependent on DL technology. LSTM is an improved version of RNN that differs from it in one important way: its architecture incorporates a memory cell at the top that allows for efficient information transmission from one instance to the next. In comparison to RNN, it can recall a large amount of information from previous states while avoiding the vanishing gradient problem [57]. By using a novel additive gradient structure that gives direct access to forgotten gate activations, LSTMs are able to tackle the vanishing gradient issue. By often changing the gates at each stage of the learning process, the network may exploit the error gradient to encourage desired behavior. A valve is used to add information to or remove it from the memory cell. The LSTM receives an input from the hidden layer of the current time instance and output from the hidden layer of the prior time instance. These two pieces of data pass via a number of network activation functions and valves before exiting at the output. The LSTM has three gates: an input gate, an output gate, and a forget gate. The forget gate and the input gate, as depicted in Figure 3, select information to be cleaned and appended to the cell state. The cell state can be updated after these two points are known. Finally, the output gate determines the network's final output [57].

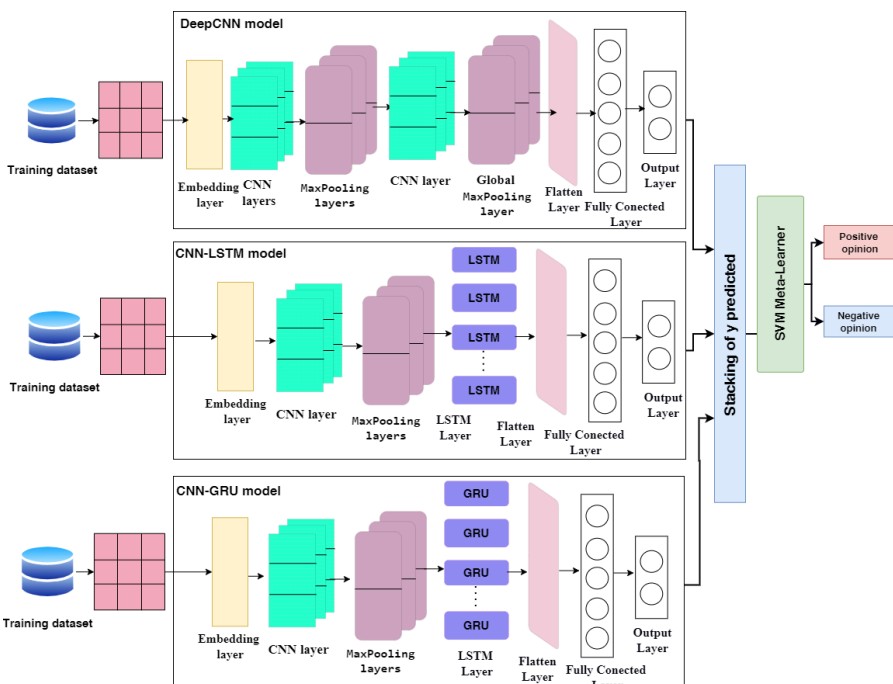

**Figure 2.** The architecture of the proposed ensemble model.

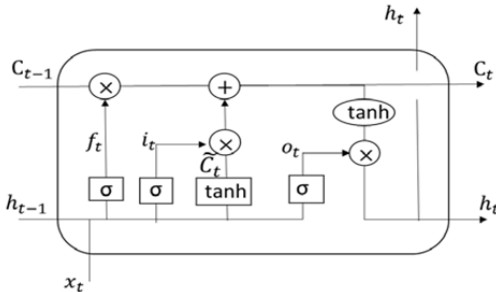

**Figure 3.** LSTM representation.

- The hybrid CNN-GRU model includes the embedding layer, CNN layers, MaxPooling layers, gated recurrent unit (GRU), fully connected layer, and output layer. The GRU model is described in detail in the following.

  GRU was designed to overcome the long-short dependence problem by removing and inflating gradients. As a result, GRU is designed to operate with sequential data that display patterns across time increments, such as time-series data. Because GRU's architecture is simpler than that of LSTM, its training speed is slightly faster than that of LSTM. The quantity of data that should be added to the next state cell is specified by the update gate. More information is transferred to the next state cell when the update gate value is larger [58]. The reset gate controls how much prior data are deleted. As a result, some information generated in the previous cell may be disregarded or forgotten as the reset gate value changes. Therefore, the update gate is responsible for ensuring that valuable memory is kept so that the next state may be passed on. This is highly beneficial since the model can select duplicating all previous data while avoiding vanishing gradients. The reset gate alters the manner in which new data are stored in previously recorded memory [59]. The GRU operation flow is depicted in Figure 4.

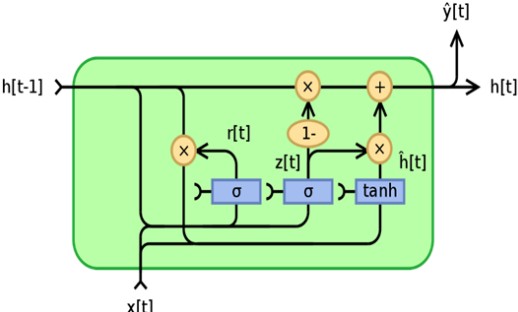

**Figure 4.** Representation for GRU [59].

*3.6. The Proposed Ensemble Model*

Instead of employing a single model, the ensemble approaches improve model accuracy by mixing numerous models. The combined models increase the outcome's accuracy considerably [60]. A stacking ensemble is an ensemble method in which a new model learns how to blend predictions from numerous existing models as best as possible. It combines predictions from a number of different trained models [61]. CNN models and hybrid models of CNN and other DL models, such as long short-term memory (LSTM), provided significant improvements in performance in sentiment analysis [4,5]. CNN uses different deep layers to extract more deep and important features from text data. LSTM has a memory state that is effective at memorizing important information in the text and understanding the meaning of the whole sentence. Therefore, we proposed an optimized heterogeneous ensemble stacking model based on the best combination of CNN, hybrid CNN-LSTM, and CNN-GRU for Arabic sentiment analysis with SVM as a meta-learner. In our work, our model is developed in many steps, as shown in Figure 2.

- The pre-trained models of DeepCNN, CNN-LSTM, and CNN-GRU that are described in Section 3.5 are loaded, and all layers of the model are frozen without the output layers.
- The output prediction of the training set for each pre-trained model are combined in the training stacking. Then, the stacking is used to train and optimize the meta-learner (SVM in our case). SVM as a meta-learner is optimized using grid search.
- The output predictions of the testing set for each pre-trained model are combined in the testing stacking. Then, the testing stacking is used to evaluate the meta-learner (SVM) using accuracy, precision, recall, f1-score and ROC.

Three datasets were divided into two parts: 80% training and 20% testing. Models were optimized using the training set. To extract features and generate feature matrices for DL models, CBOW and SkipGram word embedding were utilized. The final values of each parameter of DeepCNN, CNN-LSTM and CNN-GRU were applied with CBOW and SkipGram word embedding for each dataset, as shown in Tables 2 and 3, respectively.

**Table 2.** Values of parameters after applying DeepCNN, CNN-LSTM and CNN-GRU with CBOW.

| Models | Parameters | Values for Main-AHS Dataset | Values for Sub-AHS Dataset | Values for ASTD Dataset |
|---|---|---|---|---|
| CNN | Num_filters | [256, 128, 128] | [128, 256, 256] | [256, 128, 128] |
| | Kernel_size | [4, 5, 2] | [4, 5, 4] | [3, 5, 4] |
| | Pool_Size | [4, 5, 3] | [2, 4] | [2, 5] |
| | Dense_Unit | 300 | 150 | 500 |
| | learning_rate | 0.0012 | 0.00152 | 0.0007 |
| CNN-LSTM | Num_filters | 128 | 256 | 256 |
| | Kernel_size | 4 | 4 | 3 |
| | Pool_Size | 5 | 2 | 2 |
| | LSTM_Unit | 400 | 500 | 550 |
| | Dense_Unit | 800 | 700 | 300 |
| | learning_rate | 0.0012 | 0.0025 | 0.0046 |
| CNN-GRU | Num_filters | 256 | 64 | 128 |
| | Kernel_size | 5 | 3 | 5 |
| | Pool_Size | 2 | 3 | 2 |
| | GRU_Unit | 950 | 150 | 250 |
| | Dense_Unit | 150 | 900 | 300 |
| | learning_rate | 0.0012 | 0.0059 | 0.0021 |

**Table 3.** Values of parameters after applying DeepCNN, CNN-LSTM and CNN-GRU models with SkipGram.

| Models | Parameters | Values for Main-AHS Dataset | Values for Sub-AHS Dataset | Values for ASTD Dataset |
|---|---|---|---|---|
| CNN | Num_filters | [256, 256, 512] | [128, 128, 500] | [128, 256, 512] |
| | Kernel_size | [5, 4, 4] | [5, 5, 5] | [2, 5, 5] |
| | Pool_Size | [2, 4] | [2, 4] | [2, 5] |
| | Dense_Unit | 500 | 300 | 400 |
| | learning_rate | 0.0012 | 0.0014 | 0.00304 |
| CNN-LSTM | Num_filters | 64 | 512 | 200 |
| | Kernel_size | 5 | 5 | 2 |
| | Pool_Size | 3 | 2 | 2 |
| | LSTM_Unit | 200 | 600 | 200 |
| | Dense_Unit | 450 | 680 | 600 |
| | learning_rate | 0.00525 | 0.00051 | 0.0007 |
| CNN-GRU | Num_filters | 64 | 64 | 128 |
| | Kernel_size | 3 | 5 | 5 |
| | Pool_Size | 5 | 3 | 4 |
| | GRU_Unit | 350 | 450 | 100 |
| | Dense_Unit | 200 | 870 | 950 |
| | learning_rate | 0.0054 | 0.00142 | 0.0013 |

## 4. Experiments Results

This section describes the utilized datasets, evaluation metrics, experimental setup, and the results.

### 4.1. Datasets

We used three databases.

#### 4.1.1. Arabic Health Services Dataset (Main-AHS)

The Main-AHS dataset was collected from Twitter about healthcare services [62]. Main-AHS is classified as either 502 positive classes or 1231 classes; therefore, it is an unbalanced dataset. The Main-AHS includes two columns: text and class label. There are 2026 rows in the dataset. It does not have missing values. It is classified into 628 positive or 1398 classes; therefore, it is an unbalanced dataset. The Main-AHS dataset was divided into two parts: training and testing. The training set includes 1118 negative tweets and 502 positive tweets. The testing set includes 280 negative tweets and 126 positive tweets.

#### 4.1.2. Arabic Health Services Dataset (Sub-AHS Dataset)

Sub-AHS is a subset of the Main-AHS dataset [63]. It includes two columns: text and class label. There are 1733 rows in the dataset. It does not have missing values. It is classified into 502 positive or 1231 classes; therefore, it is an unbalanced dataset. The sub-AHS dataset was divided into two parts: training and testing. The training set includes 985 negative tweets and 401 positive tweets. The testing set includes 246 negative tweets and 101 positive tweets. The total tweets are 1732.

#### 4.1.3. Arabic Sentiment Tweets Dataset (ASTD)

The ASTD dataset was gathered from Twitter and is separated into four categories: objective, subjective negative, subjective positive, and subjective mixed [64]. In our paper, of

the selected tweets, 1642 were positive and 777 were negative; therefore, it is an unbalanced dataset. The total number of rows is 2419. It does not have missing values. The ASTD dataset was then divided into two parts: training and testing. The training set includes 1294 negative tweets and 641 positive tweets. The testing set includes 280 negative tweets and 136 positive tweets.

### 4.2. Evaluating Models

We used different measurement methods: accuracy, precision, recall, f1-score AUC, and ROC are used to evaluate the performance of the proposed model and DL and ML models. Each one is defined as follows:

Accuracy is calculated as the percentage between the correct predictions and the total number of tweets.

$$\text{Accuracy} = \frac{TP + TN}{TP + FP + TN + FN}. \tag{4}$$

Precision is calculated as the percentage of positive tweets that are rightly classified from the total number of positive tweets.

$$\text{Precision} = \frac{TP}{TP + FP} \tag{5}$$

Recall is calculated as the percentage of positive tweets that are rightly classified from the total number of tweets.

$$\text{Recall} = \frac{TP}{TP + FN} \tag{6}$$

F1-score is the weighted average of precision and recall

$$\text{F1-score} = \frac{2 \cdot \text{precision} \cdot \text{recall}}{\text{precision} + \text{recall}} \tag{7}$$

where TP stands for the amount of positively predicted sentences that were correctly made, FP for negatively predicted sentences that were incorrectly made, TN for positively anticipated negative sentences that were correctly made, and FN for positively predicted sentences that were correctly made.

Furthermore, the true positive rate (TPR) and false positive rate (FPR) are plotted against one other at different threshold levels to create the receiver operating characteristic curve (ROC) [65]. Additionally, it maps the classification findings from the most favorable to the least favorable [66]. We also computed the area under the curve (AUC). AUC, where $s_p$ is the number of positive records and $n_p$, $n_n$ are the numbers of positive and negative records, respectively, evaluates how well the model can distinguish between models [66].

$$\frac{s_p - n_p + n_{(n+1)/2}}{n_p n_n} \tag{8}$$

### 4.3. Experimental Setup

Google Colab with GPU was used to run the experiments. DeepCNN, hybrid CNN-LSTM, and hybrid CNN-GRU models were implemented using Keras and were optimized using the KerasTuner. The Scikit-learn was used to create ML models, which were then optimized via grid search. Each dataset was divided into an 80% training set and a 20% testing set. Models were optimized and trained on a training set before being tested on a testing set. The results for every model for the testing set were recorded.

Several experiments were carried out to discover the best parameters for DeepCNN, CNN-LSTM, and CNN-GRU models using KerasTuner. The datasets were divided into two parts: 80% training and 20% testing. Models were optimized using the training set. To extract features and generate feature matrices for DL models, CBOW and SkipGram word embedding were utilized.

*4.4. Results*

This section compares the performance of the proposed model to classical ML and other DL models. It also displays the results of the proposed model for the three datasets of Main-AHS, Sub-AHS, and ASTD. The results are expressed in the form of accuracy, recall, precision, f1-score, and ROC curve.

4.4.1. The Performance Results of Models for Main-AHS Dataset

This section presents the performance results of three approaches applied to the Main-AHS dataset. The first approach is the ML models; RF, DT, LR, and KNN were applied with TF-IDF and Unigram, bi-gram. The second approach is the DL models, DeepCNN, CNN-LSTM, and CNN-GRU were applied with SkipGram and CBOW word embedding. The third approach is the proposed model. Table 4 shows the values of four metrics, including accuracy, precision, recall, and f1-score of the testing results for three approaches.

For ML models, LR with unigram had the highest performance (85.96% for accuracy, 86.41% for precision, 85.96% for recall, 85.13% for f1-score), while KNN with Bi-gram had the lowest (68.97% for accuracy, 47.56% for precision, 68.97% for recall, and 56.3% for f1-score). RF with unigram registered the second-highest result (83.74% for accuracy, 84.43% for precision, 83.74% for recall, and 82.47% for f1-score).

**Table 4.** The performance results of models for the Main-AHS dataset.

| Approaches | Models | Feature Extraction Method | Testing Performance | | | |
|---|---|---|---|---|---|---|
| | | | **Accuracy** | **Precision** | **Recall** | **F1-Score** |
| ML approach | KNN | Unigram | 82.02 | 83.1 | 82.02 | 80.22 |
| | | Bi-gram | 68.97 | 47.56 | 68.97 | 56.3 |
| | DT | Unigram | 79.56 | 79.35 | 79.56 | 79.44 |
| | | Bi-gram | 77.09 | 76.25 | 77.09 | 75.25 |
| | LR | Unigram | 85.96 | 86.41 | 85.96 | 85.13 |
| | | Bi-gram | 78.33 | 77.63 | 78.33 | 76.82 |
| | NB | Unigram | 82.02 | 85.29 | 82.02 | 79.5 |
| | | Bi-gram | 82.02 | 82.38 | 82.02 | 80.57 |
| | RF | Unigram | 83.74 | 84.43 | 83.74 | 82.47 |
| | | Bi-gram | 78.08 | 78.32 | 78.08 | 75.53 |
| DL approach | Deep CNN | SkipGram | 89.41 | 89.49 | 89.41 | 89.44 |
| | CNN-LSTM | | 90.38 | 90.36 | 90.38 | 90.37 |
| | CNN-GRU | | 90.89 | 91.08 | 90.89 | 90.95 |
| | Deep CNN | CBOW | 90.64 | 90.74 | 90.64 | 90.68 |
| | CNN-LSTM | | 91.38 | 91.31 | 91.38 | 91.27 |
| | CNN-GRU | | 91.02 | 91.02 | 91.02 | 91.02 |
| The proposed ensemble model | Stacking LR | SkipGram | 91.63 | 91.56 | 91.63 | 91.57 |
| | Stacking LR | CBOW | 92.12 | 92.16 | 92.12 | 92.14 |

DL models showed an improvement in terms of accuracy, precision, recall, and f1-score when CBOW word embedding was applied. CNN-LSTM with CBOW achieved the highest performance (91.38% for accuracy, 91.31% for precision, 91.38% for recall, and 91.27% for f1-score), and it improved accuracy by 5.42%, precision by 4.9%, recall by 5.42%, and f1-score by 6.14% compared to LR with the unigram model. DeepCNN with SkipGram produced the lowest performance: 89.41% for accuracy, 89.49% for precision, 89.41% for recall, and 89.44% for f1-score).

We noticed that the classification performance of the proposed model with CBOW improved accuracy by 0.74%, precision by 0.85%, recall by 0.74%, and f1-score by 0.87% compared to CNN-LSTM with CBOW.

Additionally, Figure 5 presents ROC curve and AUC scores of ML, DL models, and the proposed model for the Main-AHS dataset. The proposed model with CBOW word embedding achieved the highest AUC scores at 91. ML models with BI-gram have the lowest AUC scores at 50, 67.421, 69.881, 67.302 for KNN, DT, LR, and RF, respectively, compared to DL models and the proposed models. The proposed model registers the second-best AUC score with SkipGram word embedding at 90.

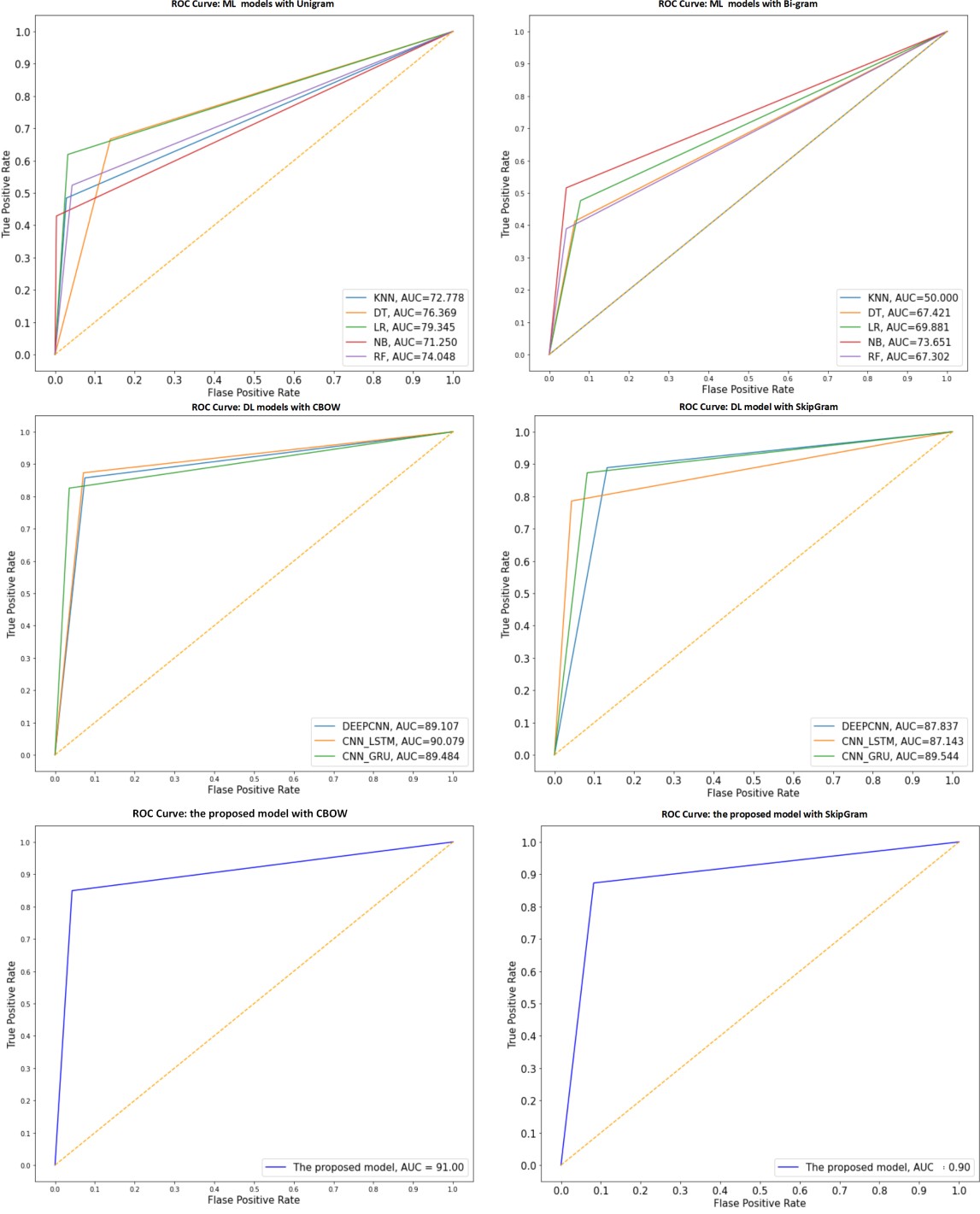

**Figure 5.** The ROC curve and AUC scores for the Main-AHS dataset.

Overall, the proposed model with CBOW word embedding achieved the highest performance compared to ML models and DL models, and all evaluation metrics are consistent: accuracy, precision, recall, f1-score, and AUC.

### 4.4.2. The Performance Results of Models for Sub-AHS Dataset

This section presents the performance results of the three approaches applied to the Sub-AHS dataset. The first approach is ML models; RF, DT, LR, and KNN were applied with TF-IDF and Unigram, bi-gram. The second approach is DL models; DeepCNN, CNN-LSTM, and CNN-GRU were applied with SkipGram and CBOW word embedding. The third approach is the proposed model. Table 5 shows the values of four metrics, including accuracy, precision, recall, and f1-score, of the testing results for the three approaches.

For ML models, LR with unigram had the highest performance (85.96% for accuracy, 86.41% for precision, 85.96% for recall, and 85.13% for f1-score), while KNN with Bi-gram had 68.97% for accuracy, 47.56% for precision, 68.97% for recall, and 56.3% for f1-score). RF with unigram registered the second-highest result (83.74% for accuracy, 84.43% for precision, 83.74% for recall, and 82.47% for f1-score).

For DL models, DeepCNN with CBOW achieved the highest performance (91.38% for accuracy, 91.31% for precision, 91.38% for recall, and 91.27% for f1-score), and it improved accuracy by 5.42%, precision by 4.9%, recall by 5.42%, and f1-score by 6.14% compared to LR with the unigram model. DeepCNN with SkipGram reached the lowest performance (89.41% for accuracy, 89.49% for precision, 89.41% for recall, and 89.44% for f1-score).

We noticed that the classification performance of the proposed model with CBOW improved accuracy by 0.74%, precision by 0.85%, recall by 0.74%, and f1-score by 0.87% compared to CNN-LSTM with CBOW.

Overall, we can see that the proposed model achieved the highest result compared to ML models and DL models.

**Table 5.** The performance results of models for the Sub-AHS dataset.

| Approaches | Models | Feature Extraction Method | Testing Performance | | | |
|---|---|---|---|---|---|---|
| | | | Accuracy | Precision | Recall | F1-Score |
| ML approach | KNN | Unigram | 79.83 | 82.32 | 79.83 | 76.31 |
| | | Bi-gram | 70.89 | 50.26 | 70.89 | 58.82 |
| | DT | Unigram | 83.0 | 82.86 | 83.0 | 82.92 |
| | | Bi-gram | 80.12 | 80.04 | 80.12 | 77.97 |
| | LR | Unigram | 87.9 | 88.29 | 87.9 | 87.19 |
| | | Bi-gram | 82.71 | 82.69 | 82.71 | 81.33 |
| | NB | Unigram | 80.69 | 84.15 | 80.69 | 77.24 |
| | | Bi-gram | 83.29 | 83.14 | 83.29 | 82.13 |
| | RF | Unigram | 86.17 | 86.96 | 86.17 | 85.06 |
| | | Bi-gram | 81.56 | 82.1 | 81.56 | 79.5 |
| DL approach | Deep CNN | SkipGram | 93.08 | 93.13 | 93.08 | 93.1 |
| | CNN-LSTM | | 93.66 | 93.61 | 93.66 | 93.6 |
| | CNN-GRU | | 93.20 | 93.30 | 93. 20 | 93.30 |
| | Deep CNN | CBOW | 94.24 | 94.21 | 94.24 | 94.22 |
| | CNN-LSTM | | 93.37 | 93.34 | 93.37 | 93.28 |
| | CNN-GRU | | 93.66 | 93.61 | 93.66 | 93.6 |
| The proposed ensemble model | Stacking SVM | SkipGram | 94.95 | 94.9 | 94.95 | 94.9 |
| | Stacking SVM | CBOW | 95.81 | 96.06 | 95.81 | 95.67 |

Additionally, Figure 6 presents ROC curve and AUC scores of ML, DL models, and the proposed model for the Sub-AHS dataset. The proposed model with CBOW and SkipGram word embedding and DEEPCNN achieved the highest AUC scores of 93. ML models with BI-gram have the lowest AUC scores of 50, 69.546, 73.507, 72.136 for KNN, DT, LR, and RF, respectively, compared to DL models and the proposed models.

Overall, the proposed model with CBOW word embedding achieved the highest performance compared to ML models and DL models, and all evaluation metrics are consistent: accuracy, precision, recall, f1-score, and AUC.

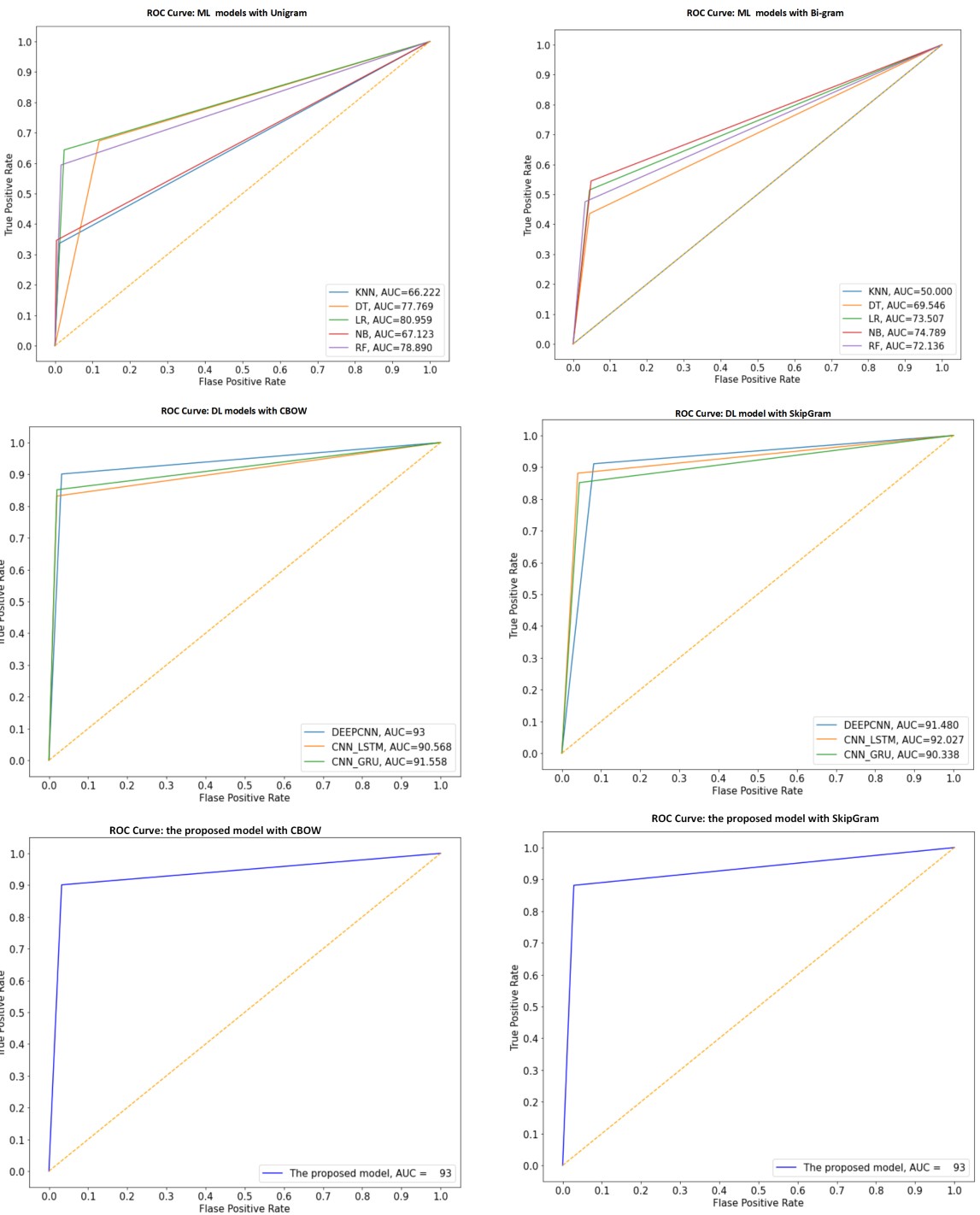

**Figure 6.** The ROC curve and AUC scores for the Sub-AHS dataset.

### 4.4.3. The Performance Results of Models for ASTD Dataset

This section presents the performance results of the three approaches applied to the ASTD dataset. The first approach is ML models; RF, DT, LR, and KNN were applied with TF-IDF and Unigram, bi-gram. The second approach is the DL models, where DeepCNN, CNN-LSTM, and CNN-GRU were applied with SkipGram and CBOW word embedding. The third approach is the proposed model. Table 6 shows the values of four metrics, including accuracy, precision, recall, and f1-score of testing results for the three approaches.

For ML models, LR with unigram had the highest performance (74.79% for accuracy, 72.65% for precision, 74.79% for recall, 70.35 % for f1-score), while DT with Unigram had the lowest (56.61% for accuracy, 63.34% for precision, 56.61% for recall, 58.69% for f1-score). NB with unigram registered the second-highest result (73.55% for accuracy, 70.86% for precision, 73.55% for recall, 69.86% for f1-score).

DL models showed an improvement in accuracy, precision, recall, and f1-score compared to ML algorithms. CNN-LSTM with SkipGram achieved the highest performance (78.51% for accuracy, 79.75% for precision, 78.51% for recall, 78.97% for f1-score), and it improved accuracy by 3.72%, precision by 7.1%, recall by 3.72%, and f1-score by 8.62% compared to LR with unigram. DeepCNN with SkipGram reached the lowest performance (71.28% for accuracy, 74.73% for precision, 71.28% for recall, 72.36% for f1-score).

We noticed that the classification performance of the proposed model improved accuracy by 2.89%, precision by 0.94%, recall by 2.89%, and f1-score by 1.19% compared to CNN-LSTM with SkipGram.

Overall, we can see that the proposed model achieved the highest result compared to ML models and DL models.

**Table 6.** The performance results of models for the ASTD dataset.

| Approaches | Models | Feature Extraction Method | Testing Performance | | | |
|---|---|---|---|---|---|---|
| | | | Accuracy | Precision | Recall | F1-Score |
| ML approach | KNN | Unigram | 71.9 | 51.7 | 71.9 | 60.15 |
| | | Bi-gram | 71.9 | 51.7 | 71.9 | 60.15 |
| | DT | Unigram | 56.61 | 63.34 | 56.61 | 58.69 |
| | | Bi-gram | 71.9 | 66.43 | 71.9 | 63.46 |
| | LR | Unigram | 74.79 | 72.65 | 74.79 | 70.35 |
| | | Bi-gram | 71.69 | 65.71 | 71.69 | 63.33 |
| | NB | Unigram | 73.55 | 70.86 | 73.55 | 69.86 |
| | | Bi-gram | 72.31 | 68.07 | 72.31 | 63.71 |
| | RF | Unigram | 64.67 | 64.91 | 64.67 | 64.79 |
| | | Bi-gram | 71.9 | 66.36 | 71.9 | 63.17 |
| DL approach | Deep CNN | SkipGram | 71.28 | 74.73 | 71.28 | 72.36 |
| | CNN-LSTM | | 78.51 | 79.75 | 78.51 | 78.97 |
| | CNN-GRU | | 77.69 | 79.5 | 77.69 | 78.29 |
| | Deep CNN | CBOW | 76.86 | 78.71 | 76.86 | 77.49 |
| | CNN-LSTM | | 78.1 | 78.41 | 78.1 | 78.24 |
| | CNN-GRU | | 78.1 | 79.35 | 78.1 | 78.56 |
| The proposed ensemble model | Stacking SVM | SkipGram | 80.17 | 79.52 | 80.17 | 79.71 |
| | Stacking SVM | CBOW | 81.4 | 80.69 | 81.4 | 80.16 |

Additionally, Figure 7 presents ROC curve and AUC scores of ML, DL models, and the proposed model for the ASTD dataset. The proposed model with CBOW word embedding

achieved the highest AUC scores at 77. ML models with Bi-gram have the lowest AUC scores of 50, 52,240, 52.096, 52.527 and 52.383 for KNN, DT, LR, NB, and RF, respectively, compared to DL models and the proposed models. The proposed model registers the best-second AUC score with SkipGram word embedding of 75.

Overall, the proposed model with CBOW word embedding achieved the highest performance compared to ML models and DL models, and all evaluation metrics are consistent: accuracy, precision, recall, f1-score, and AUC.

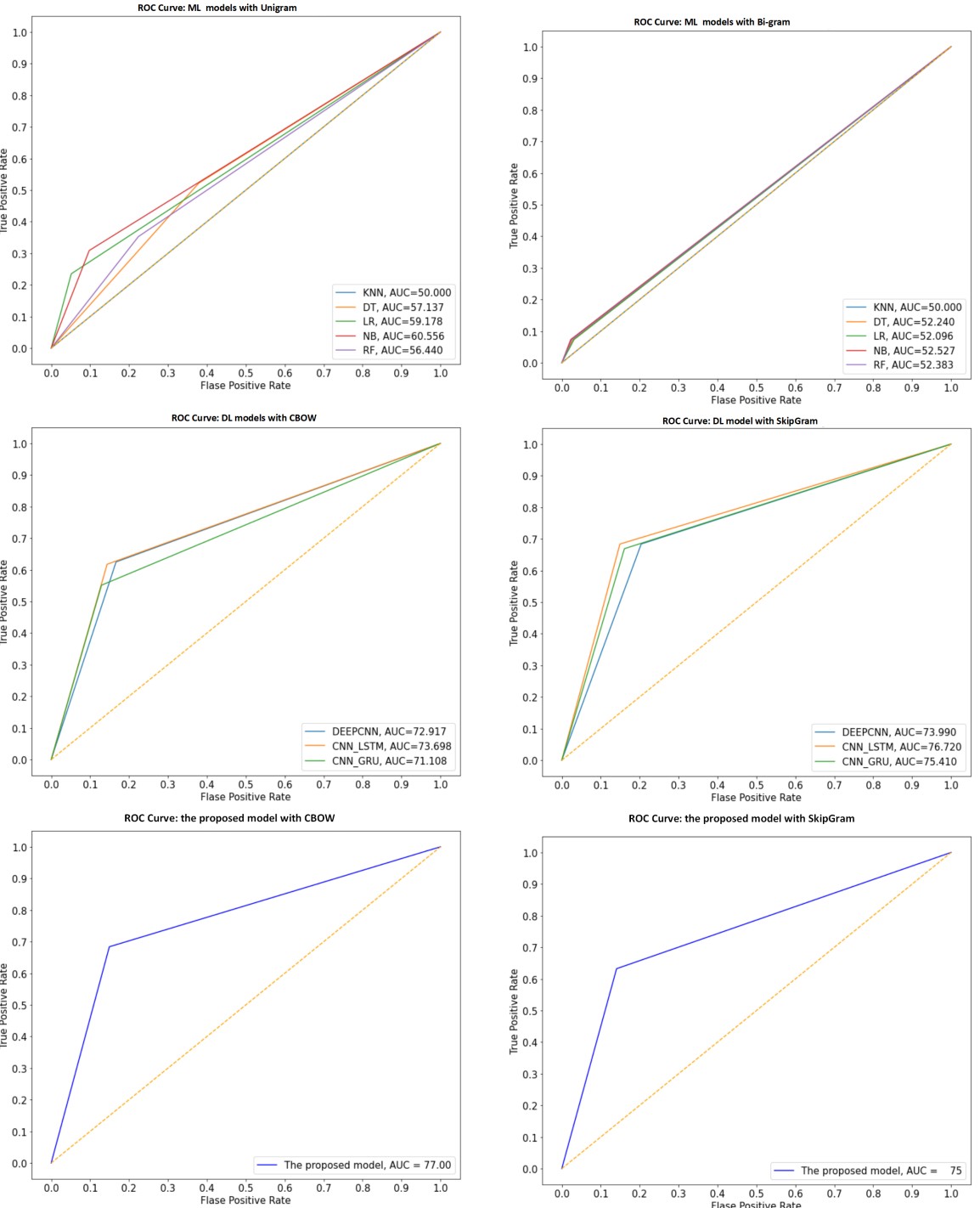

**Figure 7.** The AUC score for the ASTD dataset.

*4.5. Discussion*

Figures 8–10 show the best performing models of ML, DL and the proposed models for each of the three datasets. The best performing model from ML algorithms is the LR for the three datasets. The hybrid CNN-LSTM model achieved the best results from the optimized DL models. These models are compared with the proposed stacking ensemble model. We can see that the proposed models with CBOW for all datasets produced the highest results compared to ML and DL models. For the Main-AHS dataset, the proposed model with CBOW has obtained the highest terms of performance at accuracy = 92.12%, Precision = 91.31%, Recall = 91.38%, and f1-score = 91.27%. While LR with Unigram has the worst terms of performance at accuracy = 85.96% , Precision = 86.41%,Recall = 85.96%, and f1-score = 85.13%. For the Sub-AHS dataset, the proposed model with CBOW has obtained the highest terms of performance accuracy = 95.81%, Precision = 96.06%, Recall = 95.81%, and f1-score = 95.67%. LR with Unigram has the worst terms of performance accuracy = 87.9%, Precision = 88.29% of, Recall = 87.9% of, and f1-score = 87.19%. For the ASTD dataset, the proposed model with CBOW has obtained the highest terms of performance at accuracy = 81.4%, Precision = 80.69%, Recall = 81.4%, and f1-score = 80%. LR with Unigram has the worst terms of performance at accuracy = 74.79%, Precision = 72.65%, Recall = 74.79%, and f1-score = 70.35%.

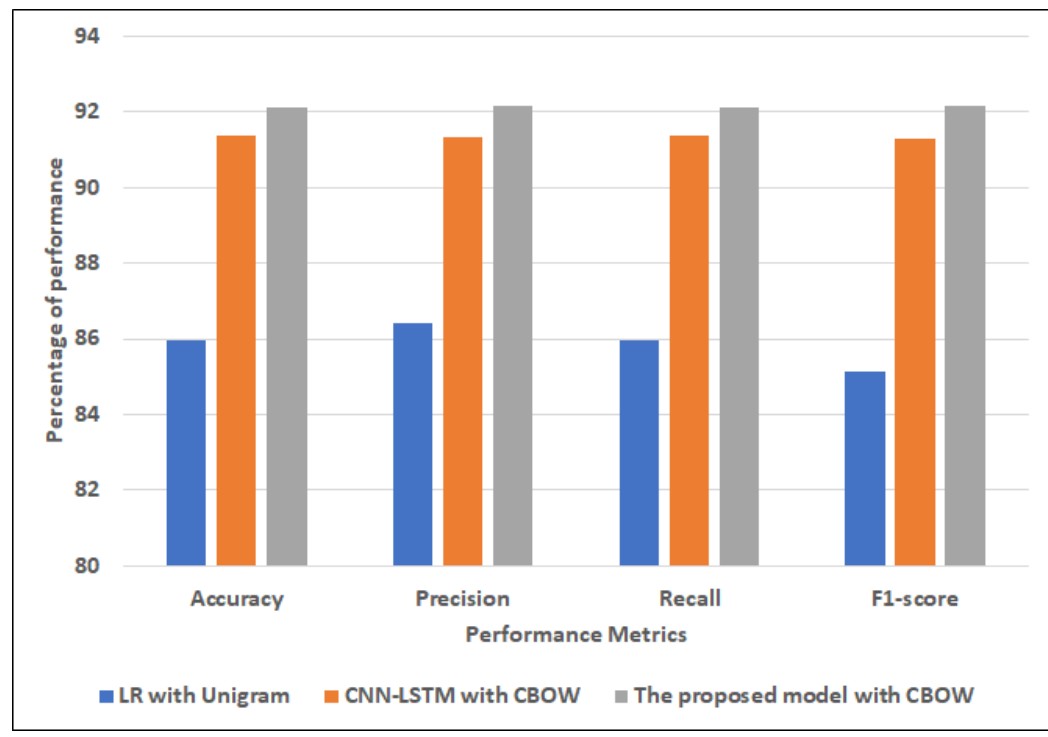

**Figure 8.** The best models for the Main-AHS dataset.

The proposed model is compared with the existing literature for the three datasets in Table 7.

Comparing the proposed model with the existing models proved that our model improved the performance of other approaches. For comparison with the authors who used the ASTD dataset in [9], the accuracy of CNN-LSTM was registered as 65.05%, and the f1-score of CNN-LSTM was registered as 64.46%. Performance in [17] for CNN-LSTM was registered as 66% for accuracy, 62% for f1-score, and 66% for recall. The accuracy of CNN-LSTM was 79.18% in [7]. In [18], the accuracy of CNN was registered as 79.07%. In [9], the authors used an ensemble model using voting based on CNN-LSTM, which was registered as 64.46% for f1-score. In [21], the Bi-LSTM was registered as 79.25% for accuracy and 76.83% for f1-score.

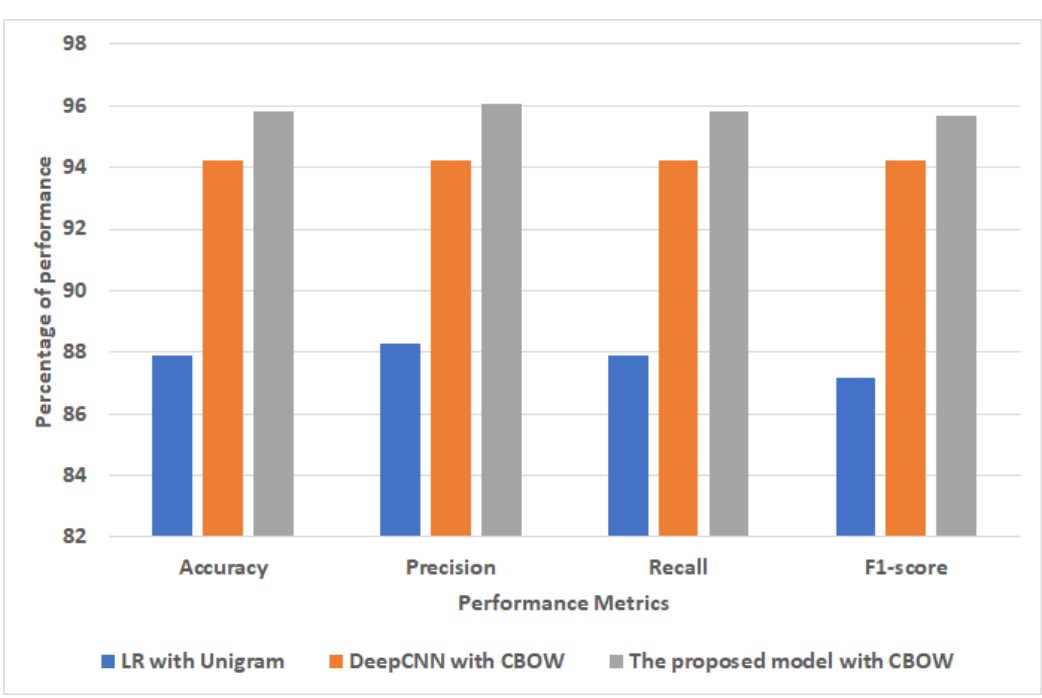

**Figure 9.** The best models for the Sub-AHS dataset.

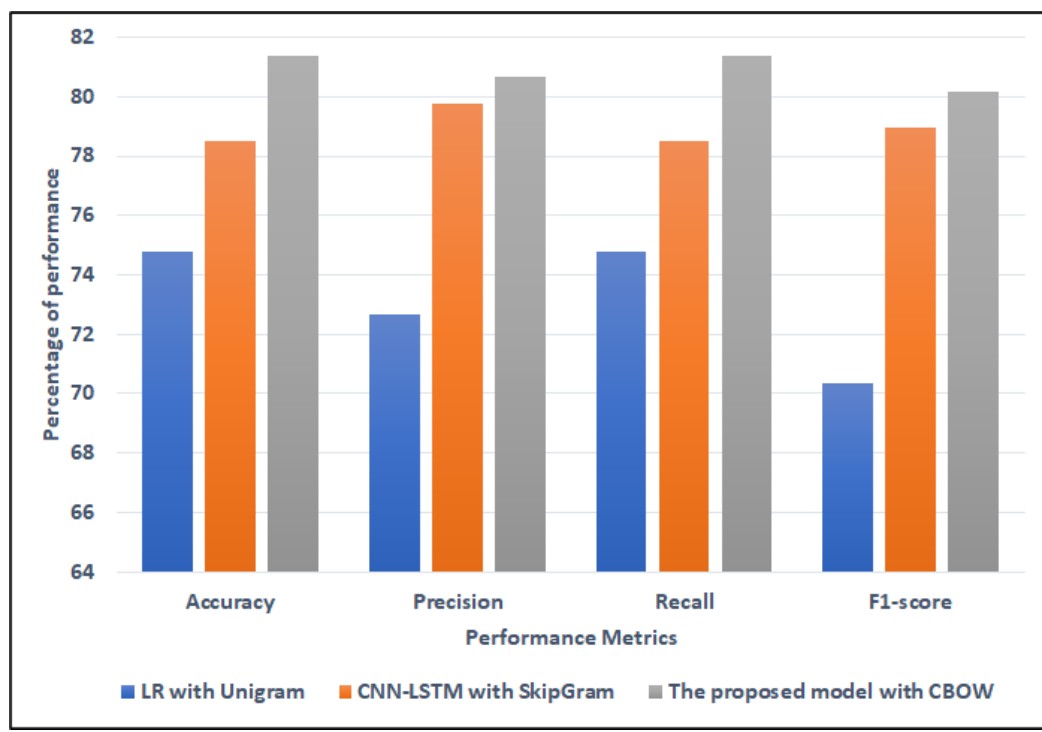

**Figure 10.** The best models for the ASTD dataset.

For Main-AHS, an accuracy of 88% was obtained in [7]. In [21], Bi-LSTM was registered with 92.61% for accuracy and 86.03% for f1-score. In [63], an accuracy of 92% was obtained. For Sub-AHS, in [63], the accuracy was registered as 95%. Overall, the proposed model achieved the highest performance compared to approaches that were used in the existing literature.

**Table 7.** Comparison of previous studies and the proposed models.

| Paper | Method | Dataset | Performance |
|---|---|---|---|
| [9] | CNN-LSTM | ASTD | 65.05% for accuracy<br>64.46% for f1-score |
| [17] | CNN-LSTM | ASTD | 66% for accuracy,<br>62% for f1-score<br>66% for recall |
| [7] | CNN-LSTM | Main-AHS | 88.1% for accuracy |
| | | ASTD | 79.18 % for accuracy |
| [18] | CNN | ASTD | 79.07% for accuracy |
| [21] | ensemble model using voting based on CNN-LSTM | ASTD | 64.46% for f1-score |
| [63] | Bi-LSTM | ASTD | 79.25% for Accuracy<br>76.83 of F1-score |
| | | Main-AHS | 92.61% for Accuracy<br>86.03% of f1-score |
| [9] | CNN | Main-AHS | 92% for accuracy |
| | | Sub-AHS | 95% for accuracy |
| The proposed model | Stacking SVM based on integrated DeepCNN, CNN-LSTM and CNN-GRU | Main-AHS | 92.12% for accuracy<br>91.31% for precision<br>91.38% of recall<br>91.27% of f1-score |
| | | Sub-AHS | 95.81% for accuracy<br>96.06% for precision<br>95.81% for recall<br>95.67% for f1-score |
| | | ASTD | 81.4% for accuracy<br>80.69% of precision<br>81.4% for recall<br>80% for f1-score |

## 5. Conclusions

This study addresses the problem of sentiment analysis for Arabic text. The performance of the Arabic sentiment analysis system was examined in relation to the use of ensemble stacking models based on CNN, a hybrid CNN-LSTM model, and a hybrid CNN-GRU model. Additionally, the ensemble stacking models have made important contributions to increasing NLP accuracy. In order to improve the model's performance in forecasting Arabic sentiment analysis, we suggested an optimal ensemble staking model that includes three pre-trained models: deep layers of CNN, hybrid CNN-LSTM, and hybrid CNN-GRU, together with a meta-learner SVM. Different layers are included in the DeepCNN model: the flatten layer, fully connected layer, output layer, two MaxPooling layers, three CNN layers, and the global MaxPooling layer. The embedding layer, CNN and MaxPooling layers, long short-term memory (LSTM), fully linked layer, and output layer are all components of the hybrid CNN-LSTM model. The embedding layer, CNN and MaxPooling layers, the gated recurrent unit (GRU), the fully linked layer, and the output layer are all components of the hybrid CNN-GRU model. To extract features for DL models, CBOW and the skip-gram models with 300 dimensions word embedding were utilized. The performance of the proposed model is evaluated against DeepCNN, hybrid CNN-LSTM, hybrid CNN-GRU, and conventional ML algorithms. The findings demonstrate that, when compared to other models, the suggested ensemble model has the best performance for each dataset. The proposed model with CBOW word embedding has the highest accuracy of 92.12%, 95.81%, and 81.4% for Main-AHS, Sub-AHS, and ASTD, respectively.

In future work, the word sense in Arabic is critical and expected to improve performance [67], and we will consider this limitation in the future.

**Author Contributions:** Methodology, H.S.; Validation, H.S.; Visualization, H.S. and S.M.; Writing—review & editing, H.S., S.M., S.E.-S., L.A.G. and A.O.A. All authors have read and agreed to the published version of the manuscript.

**Funding:** This research received no external funding.

**Institutional Review Board Statement:** Not applicable.

**Informed Consent Statement:** Not applicable.

**Data Availability Statement:** All datasets used to support the findings of this study are available from the direct link in the dataset citations.

**Acknowledgments:** Princess Nourah bint Abdulrahman University Researchers Supporting Project number (PNURSP2022R178), Princess Nourah bint Abdulrahman University, Riyadh, Saudi Arabia.

**Conflicts of Interest:** All authors declare that they have no conflict of interest.

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
