# Peer review of "Enhanced Arabic Sentiment Analysis Using a Novel Stacking Ensemble of Hybrid and Deep Learning Models"

_applsci, doi:10.3390/app12188967_

Round 1
Reviewer 1 Report
The paper shows a new approach to sentiment analysis for the Arabic language.
The authors proved an enhanced performance of the proposed approach.
However, some enhancements should be done:
1) references: why the reference list starts from 5? it should be corrected
2) A description of the data is missing. How many records are there in the datasets? is it a balanced dataset?
3) You are including both accuracies, ROC, and F1-score in the results. Why? A better discussion should be included to show which one should be considered for the decision on performance.
4) Another related work published in Applied Sciences journal: https://www.mdpi.com/2076-3417/11/11/4768
How your work is different than it? or how your work can contribute to it?
5) Did you consider the word sense? If not, mention this in the scope of your research. You can find details about word sense in this paper: https://wires.onlinelibrary.wiley.com/doi/abs/10.1002/widm.1447
Reviewer 2 Report
This paper proposes the analysis of sentiment over Arabic text by combining the CNN model known as powerful in prediction and the hybrid deep learning model with SVM as a meta-learner. The related works section presents a good overview of existing attempts, espacially using DL models. The paper has the merit to be accepted after addressing and clarifying some minor points, particularly:
- The description of the methodlogy requires some examples to illustrate the novelty of the proposed approach. Indeed, in the pre-processing step, for instance, what kind of stop words are deleted? some of them after being deleted can inverse the sentiment resluting in a false result.
- When using a third party software such as AraVec word embedding it'll be interesting to show its accuracy and shortecomings in order to undertand at what extent it can impact the experiment results.
- The optimized heterogeneous ensemble stacking model based on the best combination of CNN, hybrid CNN-LSTM, and CNN-GRU merits to be presented at a performance level based on previous works.
- For the sake of better readability, Table 5 and 6 and Figure 6 should be summarized to show in one side the improvement registered (in terms of percentage) with the proposed model in comparison to the ML and DL models
- The used dataset has been cited with a link. In the same direction, it'll be important for research reproduction to make available online the developed material for other researchers in Arabic SA
- Figure 8 and 9 have to be placed before references
